# Dual-Modality Molecular Imaging of Tumor via Quantum Dots-Liposome–Microbubble Complexes

**DOI:** 10.3390/pharmaceutics14112510

**Published:** 2022-11-18

**Authors:** Jieqiong Wang, Yuanyuan Wang, Jie Jia, Chenxing Liu, Dong Ni, Litao Sun, Zhijie Guo

**Affiliations:** 1Department of Rehabilitation Medicine, Huashan Hospital, Fudan University, Shanghai 201206, China; 2Center for Cell and Gene Circuit Design, Shenzhen Institute of Synthetic Biology, Shenzhen Institutes of Advanced Technology, Chinese Academy of Sciences, Shenzhen 518055, China; 3Medical Ultrasound Image Computing (MUSIC) Laboratory, Shenzhen University, Shenzhen 518055, China; 4Department of Ultrasound Medicine, Zhejiang Provincial People’s Hospital (Hangzhou Medical College Affiliated People’s Hospital), Hangzhou 310014, China; 5Department of Ultrasound, Shenzhen Bao’an Chinese Medicine Hospital, Guangzhou University of Chinese Medicine, Shenzhen 518133, China

**Keywords:** dual-modality imaging, liposome–microbubble complexes, ultrasound molecular imaging, fluorescent imaging, iRGD, quantum dots

## Abstract

Molecular imaging has demonstrated promise for evaluating the expression levels of biomarkers for the early prediction of tumor progression and metastasis. However, most of the commonly used molecular imaging modalities are relatively single and have difficulties imaging complex biological processes. Here, we fabricated αvβ3-integrin-targeted quantum-dots-loaded liposome–microbubble (iRGD-QDLM) complexes that combined ultrasound imaging with optical imaging. The resulting iRGD-QDLM has excellent binding capability to 4T1 breast cancer cells. Ultrasound molecular imaging of 4T1 tumors demonstrated that significantly enhanced ultrasound molecular signals could be observed in comparison with non-targeted QDLM. Importantly, our study also suggested that iRGD-QDL on the surface of microbubbles could be delivered into a tumor by ultrasound-mediated microbubble destruction and adhered to αvβ3 integrin on breast cancer cells, achieving transvascular fluorescent imaging. Our study provides a novel approach to dual-modality molecular imaging of αvβ3 integrin in the tumor tissue.

## 1. Introduction

Breast cancer has now overtaken lung cancer as the world’s mostly commonly diagnosed cancer among the female population worldwide [1]. Metastasis accounts for the majority of deaths from breast cancer and remains a major health problem around the world [2]. Adequate prediction of metastatic occurrence will be critical for developing therapeutic interventions to combat breast cancer. Actually, breast cancer metastasis is a complex, multistep biological process that involves a cascade of events including angiogenesis, progression, extravasation, and colonization [3]. Specifically, angiogenesis is critical for the growth, metastasis, and survival of breast cancer [4,5]. Accompanied by tumor angiogenesis, cell surface receptors in endothelial cells and breast cancer cells exhibit drastic changes and are overexpressed for modulating cell adhesion, invasion, and metastasis. Among these, αvβ3 integrin is highly expressed in breast and angiogenic endothelium [6]. It is possible to gain a further understanding of tumor angiogenesis via various imaging modalities, due to an abundance of integrins on the tumorous vascular cells that provides molecular targets for delivery of imaging probes. Molecular imaging of these biomarkers with different modalities is well-developed and tracks longitudinal responses to anti-angiogenesis therapy [7,8].

To date, various in vivo imaging modalities have provided integrated information on specific molecules of interest within the cells of living subjects [9,10,11]. Among these, ultrasound molecular imaging (UMI), which relies on targeted microbubbles selectively adhering to a ligand-specific markers, has been successfully applied to the early diagnosis and dynamic monitoring of various diseases (e.g., tumors, inflammation) [12,13]. It naturally inherits many advantages of ultrasound imaging including non-invasiveness, real-time feedback, relative tolerability, and high spatial and temporal resolution. More importantly, ultrasound-stimulated microbubbles can promote the delivery of imaging probes into the tumor, achieving transvascular imaging for detecting the molecular events [14]. Despite these considerable advantages, UMI suffers from some challenges. One is that microbubbles are large and are limited to intravasculature molecular targets, and, at high US frequencies, numerous microbubbles are damaged [15]. Interestingly, optical imaging has high sensitivity and subcellular resolution for probe detection. Using specific near-infrared fluorophores, optical imaging enables disease-specific morphologic and biochemical processes beyond intravascular imaging [16]. However, it has limited tissue penetration. Therefore, dual-modality imaging that integrates ultrasound and optical imaging techniques may compensate for each other’s limitations to some degree [17].

Here, we report the quantum-dots-liposome–microbubble complexes through conjugating quantum dots (QD)-loaded iRGD-liposomes (iRGD-QDL) to the lipid shell of MBs via avidin–biotin linkage (iRGD-QDLM). Thanks to the high affinity and specificity of binding to αvβ3 integrin, the iRGD peptide (sequence CRGDKGPDC) is a widely used ligand for the design of molecular probes to assess the expression levels of αvβ3 integrin for the early detection of tumor angiogenesis [18]. In addition, quantum dots were used for fluorescent probes due to their high molar extinction coefficients with broad excitation spectra and size-tuneable, thermal, and photo-chemical stability. [19] To distinguish the ultrasound signal of MBs that had adhered to αvβ3 integrin from the signal of freely circulating iRGD-QDLM, a destruction/replenishment approach was implemented. Firstly, iRGD-QDLM was intravenously administered into the tumor-bearing mice in one bolus injection. After binding to αvβ3 integrin for several minutes, iRGD-QDLM would be destructed by a high-power destructive pulse. Ultrasound contrast signals before and after destruction in the tumor were collected. The ultrasound molecular imaging signals of iRGD-QDLM adhered to αvβ3 integrin could be calculated by deducting the post-burst contrast signals from the pre-burst contrast signals. In addition, the iRGD-QDL attached to αvβ3 integrin in endothelial cells was presented by fluorescent imaging. iRGD-QDL could be delivered into tumor by ultrasound-mediated microbubble destruction and adhered to αvβ3 integrin in breast cancer cells, achieving transvascular fluorescent imaging (Figure 1).

## 2. Materials and Methods

### 2.1. Materials

The 1,2-Distearoyl-sn-glycero-3-phosphatidylcholine (DSPC), 1,2-distearoyl-sn-glycero-3-phosphoethanolamine-N-[methoxy(polyethyleneglycol)-2000] (DSPE-PEG2000), 1,2-distearoyl-sn-glycero-3-phosphoethanolamine-N-[biotinyl (polyethylene glycol)-2000] (DSPE-PEG2000-Biotin), 1,2-dipalmitoyl-sn-glycero-3-phosphocholine (DPPC), and cholesterol were purchased from Avanti Polar Lipids Inc. (Alabaster, AL, USA). The iRGD nonapeptide (CRGDKGPDC) with a cysteine at its N-terminus was synthesized by GL Biochem (Shanghai, China). QD605 were obtained from Wuhan Jiayuan Quantum Dots Co., Ltd. (Wuhan, China). Avidin, mice anti-mouse CD31 antibodies, and goat anti-mice secondary antibodies were obtained from Sigma-Aldrich (St. Louis, MO, USA). Perfluoropropane (C_3_F_8_) was purchased from FluoroMed. Breast cancer 4T1 cell line was purchased from the American Type Culture Collection. The 6–8 weeks old female BALB/c mice (~20 g) were obtained from Guangdong Medical Experimental Animal Center (Guangzhou, China).

### 2.2. Preparation of Biotinylated MBs and Biotinylated QD-Liposomes

Biotinylated MBs were fabricated by thin-film hydration method. Briefly, appropriate proportions of DSPC, DSPE-PEG2000, and DSPE-PEG2000-Biotins (molar ratio, 90:5:5) were dissolved in 1 mL chloroform. Chloroform and phospholipids were swirled under nitrogen flow, and the solvent was completely volatilized by vacuum filtration at room temperature for 4 h. The dried phospholipid thin films were then hydrated at 60 °C with 5 mL of hydration solution (pH 7.4) consisting of 80% Tris (100 mM), 10% glycerol, and 10% propyleneglycol (*v*/*v*), followed by addition of perfluoropropane. Finally, the admixtures were oscillated for 45 s in a mechanical oscillator for obtaining biotinylated MBs. The preparation of biotinylated QD-liposomes was the same as that of biotinylated microbubbles. In brief, an admixture solution consisting of DPPC:cholesterol:DSPE-PEG2000-Biotin:iRGD-DSPE-PEG 2000:QD, at a molar ratio of 60:25:4:1:10, was prepared as described above. The solution was then extruded through a polycarbonate carbonate filter (200 nm, Avanti Polar Lipids, Alabaster, AL, USA), and iRGD-QD-liposomes were finally obtained. QD-liposomes were also fabricated as described above.

### 2.3. Conjugation of Biotinylated Liposomes to Avidin-MBs

Biotinylated MBs were washed three times with phosphate buffered sodium (PBS) solution using differential centrifugation at 400× *g* for 3 min to remove free lipids. Avidin (final concentration 0.3 mg/mL) was then added to the washed MB dispersion (2 × 10^8^ particles/mL). The avidin-MBs were washed three times to remove unreacted avidin and then incubated at room temperature with 1 mL iRGD-QD-liposomes for another 15 min. iRGD-QD-liposomes–microbubbles (iRGD-QDLM) were washed 3–4 more times to remove free iRGD-QD-liposomes. QD-liposomes–microbubbles (QDLM) were prepared in the same way.

### 2.4. Characterization of iRGD-QDLM and QDLM

iRGD-QDLM suspension (50 μL, 1 × 10^7^ particles/mL) was confirmed on a microscope slide with an inverted fluorescence microscope (Leica DMI3000B, Wetzlar, Germany). The concentration, particle size, and distribution of iRGD-QDLM and QDLM were determined using an Accusizer 780 optical particle size analyzer with a 0.5 μm in diameter lower detection limit (Accusizer 780; Particle Sizing Systems, Santa Barbara, CA, USA).

### 2.5. In Vitro Cell Binding

The 4T1 breast cancer cells were seeded at 2 × 10^4^ cells per well in 24-well plates and cultured overnight. iRGD-QDLM or QDLM (1 × 10^8^ particles/mL) were co-incubated with 4T1 breast cancer cells for 5 min. Subsequently, free MBs were removed by washing with PBS for three times. Images were acquired using light microscope (Leica DMI3000B, Wetzlar, Germany). The number of iRGD-QDLMs or QDLMs bound to the cells were counted by Image J.

### 2.6. In Vivo Ultrasound Molecular and Fluorescent Imaging

All animal experiments were approved by the management group of experimental animal nursing institutions of the Technical Research Institute of the Shenzhen Institute of Advanced Technology, Chinese Academy of Sciences (approval number: SIAT-IACUC-210303-HCS-YF-A1693; date of approval: 10 September 2021). Ultrasound molecular imaging was implemented using high-resolution ultrasound imaging system equipped with a 25-MHz high-frequency transducer (Vevo 2100; VisualSonics, Inc., Toronto, ON, Canada). All imaging parameters (grain: 35 dB; transmit power: 20%; depth: 9 mm) were set throughout the imaging process. Tumor volume (V) was calculated according to the formula: V = ((length (mm) × width ^2^ (mm)) × 0.5). When tumor volume was 150.3 ± 33.1 mm^3^, tumor-bearing mice on a heating pad were anesthetized with 2% isoflurane. To distinguish the ultrasound signal of MBs that had adhered to αvβ3 integrin from the signal of freely circulating MBs, a destruction/replenishment approach was implemented. In brief, after intravenous administration of 5 × 10^7^ iRGD-QDLM (125 μL) and waiting for 4 min prior to imaging, approximately 100 ultrasonographic frames of the tumor tissue were acquired at a temporal resolution of 10 s. Subsequently, a high-power destructive pulse was applied to destroy these MBs for one second. After the destruction procedure, another sequence of 500 frames were immediately captured to conform the degree of microbubble replenishment. All ultrasound image sequences were analyzed and quantified in tumor regions of interest using Vevo 2100 analytical software. The ultrasound molecular imaging produced by the adherent microbubbles was finally determined by the pre-destruction signals subtracting the post-destruction imaging signals, which were automatically expressed as green code overlays on the B-mode ultrasound images.

To determine the fluorescent signal produced by the adherent liposome, the fluorescent images of mice were obtained using an IVIS small animal imaging system (IVIS Spectrum, Caliper Corporation, Newton, MA, USA,) after ultrasound molecular imaging. Images were captured at 1, 6, 12, 18, and 24 h after ultrasound molecular imaging. Imaging data were analyzed and quantified using IVIS Living Image software.

### 2.7. Immunohistochemistry

Immediately after fluorescent imaging, the mice were euthanized, and their tumor tissues were harvested for immunohistochemical assays. Dissected tumor samples were covered with Tissue-Tek (Sakura), frozen in liquid nitrogen, and then serially cross-sectioned into 10 μm thick slices. Frozen sections were stained with an anti-CD31 antibody (green). Representative fluorescent images of the microvessel (CD31: green) and liposome (quantum dots: red) were captured using a confocal microscope (Leica DMI3000B, Wetzlar, Germany).

### 2.8. Statistical Analysis

Statistical analysis was carried out with GraphPad Prism 7 (GraphPad Software Inc., San Diego, CA, USA). Data were presented as mean ± standard deviations. Two groups of data were analyzed by two-tailed t-test. Multiple groups of data were analyzed by one-way ANOVA, and then differences among means were analyzed by the Student–Newman–Keuls test. Differences were considered significant at *p* < 0.05.

## 3. Results

### 3.1. Fabrication and Characterization of iRGD-QDLM

iRGD-QD-liposomes were fabricated by a thin-film hydration method. Subsequently, iRGD-QD-liposomes were attached to the surface of biotinylated MBs through a biotin–avidin linkage (iRGD-QDLM) (Figure 2a). A similar particle size and distribution for iRGD-QDLM and QDLM were observed, with the mean particle size at 1.53 ± 0.91 μm (Figure 2b). Red fluorescence at 605 nm (QD605) could be observed on the surface of MBs, confirming the successful conjugation of targeted complexes (Figure 2c).

### 3.2. Binding Specificity of iRGD-QDLM

To investigate the targeted binding capability of iRGD-QDLM to the αvβ3 integrin, iRGD-QDLM was incubated with 4T1 breast cancer cells (Figure 3a). Figure 3b demonstrated that large amounts of iRGD-QDLM were attached to the surface of 4T1 breast cancer cells. In contrast, few QDLMs were observed on the surface of 4T1 breast cancer cells (Figure 3c). Quantitative analysis revealed that the number of the iRGD-QDLMs adhered to the cells were 6.1-fold more than that of the QDLMs (Figure 3d).

### 3.3. In Vivo Contrast-Enhanced Ultrasound Molecular Imaging

We evaluated the potential of iRGD-QDLM to target tumor vessels by using a destruction/replenishment approach on the 4T1 breast cancer tumor model (Figure 4a). Representative targeted, contrast-enhanced ultrasound images of 4T1 tumor xenografts are shown in Figure 3b. Typically, ultrasound molecular signals from the control MBs (QDLM) were hardly detected in the tumors. In contrast, strong ultrasound molecular signals (green) were observed in the tumor when the targeted MBs (iRGD-QDLM) were used. Quantitative analysis of the contrast-enhanced images demonstrated that the mean signal intensity from iRGD-QDLM (28.10 ± 2.119) was significantly higher than that of QDLM (2.4 ± 1.0) (Figure 4c,d).

### 3.4. In Vivo Transvascular Fluorescent Imaging

Given that sonication can increase the vascular and tissue permeability to deliver liposomes into a tumor, the transvascular imaging of iRGD-QDLM binding tumor cells was assessed after ultrasound molecular imaging. Chronological changes in fluorescence intensity in the tumors of three xenografted mice were presented in Figure 5a. Obvious fluorescence can be observed in the tumor at 1 h post-injection for the mice receiving iRGD-QDLM + US, though not for the mice receiving QDLM + US, iRGD-QDLM or QDLM without US. The fluorescent intensity of the tumors peaked at 6 h for all four groups, then decreased with time, but the tumors that received iRGD-QDLM + US always possessed the strongest fluorescent signals (Figure 5b). Interestingly, the mean intensity in the QDLM + US group was significantly higher than that of the mice receiving iRGD-QDLM at 24 h. This can be contributed to the following two reasons: one is that liposome–microbubble complexes in combination with ultrasound-mediated microbubble destruction improved QD605 delivery into the tumor, while the other is that iRGD could increase the vascular and tissue permeability in a tumor-specific manner. To further validate the fluorescent signals from the tumor transvascular region, the tumors were dissected and subsequently analyzed for CD31 immunostaining. The results indicated that QD605 did not colocalize with CD31, confirming the presence of iRGD-QD within the 4T1 tumors (Figure 5c). Similarly, iRGD-QDLM + US revealed higher levels of fluorescent intensity than any other group, suggesting that iRGD-QDLM triggered by US released iRGD-QD into the tumor’s interstitial space. These data demonstrated the transvascular fluorescent imaging capabilities of iRGD-QDLM combined with an ultrasound.

## 4. Discussion

Tumor invasion and metastasis accounts for the majority of deaths from most cancers. Adequate detection of metastasis-related biomarkers can much better reflect the true status of tumors, thus providing suitable strategies for cancer treatment and management. Molecular imaging technologies provide noninvasive strategies for the visualization, characterization, and quantification of biologic processes at the cellular and subcellular levels. They have been used for diagnosis of tumors metastasis in pre-clinical and clinical models [20,21]. However, at present, most of the commonly used molecular imaging modalities are relatively single. In contrast, multimodal imaging provides additional spatial or molecular information [22]. In this study, we fabricated αvβ3-integrin-targeted liposome–microbubbles complexes that simultaneously combined the ultrasound imaging with optical imaging and developed a destruction/replenishment approach to achieve transvascular fluorescent imaging.

In ultrasound molecular imaging, ligand-modified MBs were utilized to specifically bind to biomarkers overexpressed in the surface of tumorous vasculature. Previous researchers prepared iRGD-lipopeptide-based MBs (iRGD-MBs) for imaging and assessing the process of tumor angiogenesis in a preclinical study [12]. Although iRGD-MBs allowed for simple fabrication and future clinical translation, they had the vascular confinement of imaging and a low sensitivity in ligand detection due to the numerous microbubbles vulnerable to ultrasonic destructions at high US frequencies. To compensate for this limitation, we conjugated iRGD-modified quantum-dots-loaded liposomes to the surface of MBs via biotin–avidin linkage, giving this system great potential for both targeted dual-modality imaging methods. Our results from the cell affinity experiments demonstrated that iRGD-QDLM can bind with vascular endothelial cells and breast cancer cells (Figure 3), proving the specific attachment between iRGD-QDLM and breast cancer cells. In addition, our molecular sonograms revealed that the ultrasound imaging signal using iRGD-QDLM was significantly higher than that using non-targeted MBs in the 4T1 tumor model (Figure 4), indicating an effective retention of iRGD-QDLM at the tumor site. These data clearly showed iRGD-QDLM still possessed ultrasound molecular imaging of tumor angiogenesis.

The successful prediction of metastasis occurrence needs non-invasive imaging approaches to monitor biological processes of the tumor itself. Advances in bioluminescence imaging provide a strategy for in vivo visualizing of dynamically tumor-associated properties. Previous studies showed that iRGD-modified liposome–microbubble complexes can be triggered by ultrasound for delivering liposome into a tumor [23]. In the current study, we employed iRGD-QDL to monitor metastasis-related gene expression on the surface of tumorous cells. As expected, iRGD-QDLM can be destroyed by pulse ultrasound, and iRGD-QDL were delivered into the inner tumor, to achieve transvascular fluorescent imaging (Figure 5). The co-localization of the CD31 proteins expressed on the endothelial cell membrane is widely considered as a marker of angiogenesis, and iRGD-QD indicated the presence of iRGD-QD within the 4T1 tumors. Compared with the molecular-imaging modalities used in clinical practice (MRI, CT, PET, and SPECT), our dual-modality imaging integrates the advantages of the non-invasiveness and relative tolerability of the ultrasound imaging and the properties of the high sensibility of optical imaging. Actually, the clinical imaging modalities suffer from major limitations including adverse effects to hazardous ionizing radiation (CT, PET, and SPECT) and intrinsically limited spatial and poor temporal resolutions (CT, MRI, PET, and SPECT). More importantly, novel dual-modality imaging achieved transvascular fluorescent imaging via ultrasound targeted destruction technology. Furthermore, it should be pointed out that there are some limitations to the present study. For example, some non-specific interference signals from the non-targeted liposome into the tumor affect the evaluation of biomarkers’ imaging on the surface of tumor cells, due to enhanced permeability and the retention effect. Although we can eliminate an interfering signal by advanced imaging processing methods, it is desired to develop a novel probe, such as an aggregation-induced emission fluorescent probe, for predicting and monitoring the biological process of tumor metastasis [24].

## 5. Conclusions

We have developed iRGD-lipopeptide-based αvβ3-integrin-targeted quantum-dots-liposome–microbubble complexes that allows for dual-modality imaging of tumor-metastasis-related biomarkers and future clinical translation. Our study suggests that iRGD-QDLM can be used for ultrasound molecular imaging of the tumor’s angiogenesis and transvascular region of the metastasis-related biomarkers on the surface of tumor cells in a preclinical study.

## Figures and Tables

**Figure 1 pharmaceutics-14-02510-f001:**
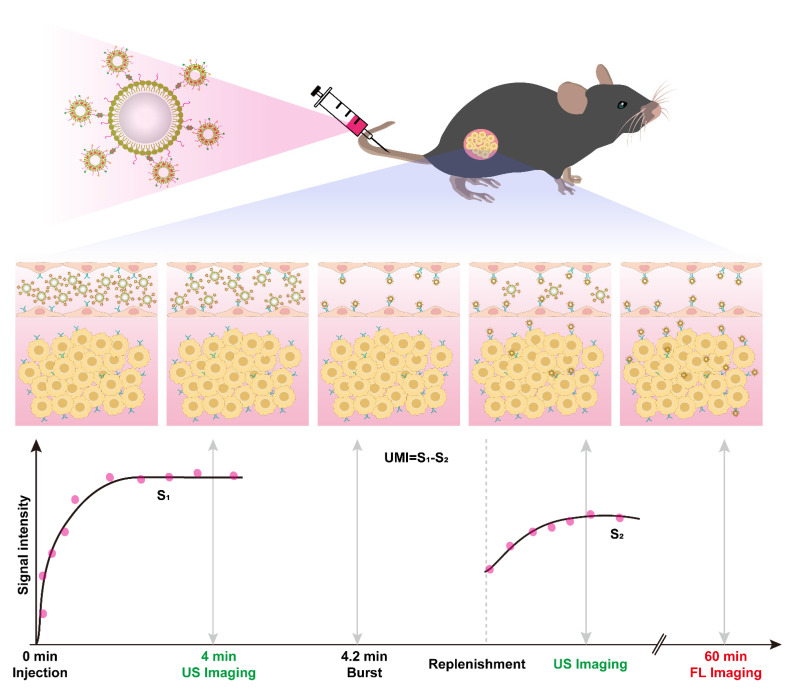
Schematic diagram of dual-modality imaging of tumor via quantum-dots-liposome–microbubble complexes (iRGD-QDLM). The αvβ3-targeting iRGD-QDLM was injected via the tail vein of mice. After allowing 4 min for these targeting MBs to bind with their receptors, a high intensity acoustic burst was used to collapse the iRGD-QDLM. The contrast signals of the binding iRGD-QDLM, which represents the molecular imaging signals for αvβ3 receptors, can be calculated by subtracting the post-burst contrast/replenishment signals (S2) from the 1st pre-burst contrast signals (S1). iRGD-QDL attached to αvβ3 integrin in endothelial cells can be detected by fluorescent imaging. Moreover, they can be delivered into tumor by ultrasound-mediated microbubble destruction and adhered to αvβ3 integrin in breast cancer cells, achieving transvascular fluorescent imaging.

**Figure 2 pharmaceutics-14-02510-f002:**
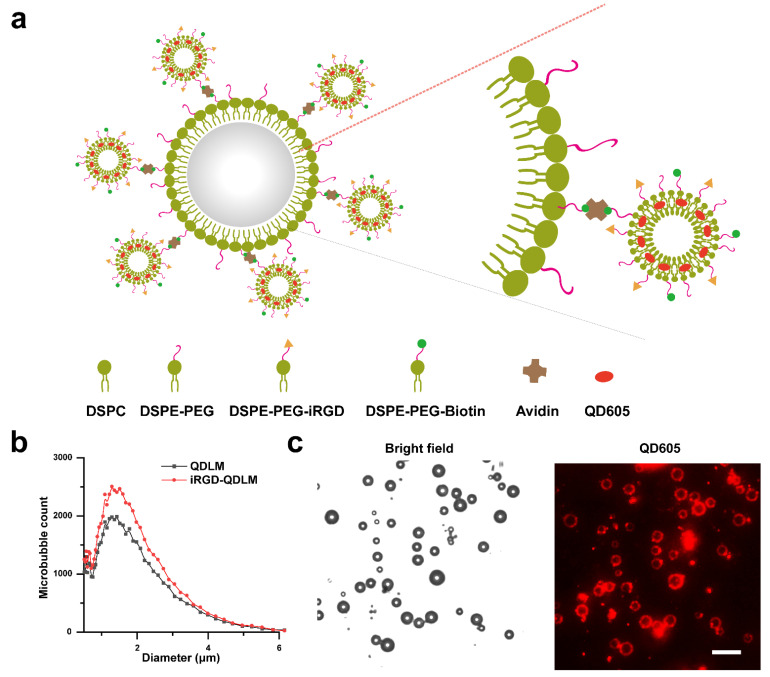
Preparation and characterization of iRGD-QDLM. (**a**) Schematic diagram of iRGD-QDLM. Biotinylated liposomes containing QD are adhered to the surface of biotinylated MBs via biotin–avidin system. (**b**) Size distribution of iRGD-QDLM and QDLM. (**c**) Light micrograph of iRGD-QDLM (scale bar: 10 μm), revealing good dispersity and QD-liposomes attached to the surface of MBs.

**Figure 3 pharmaceutics-14-02510-f003:**
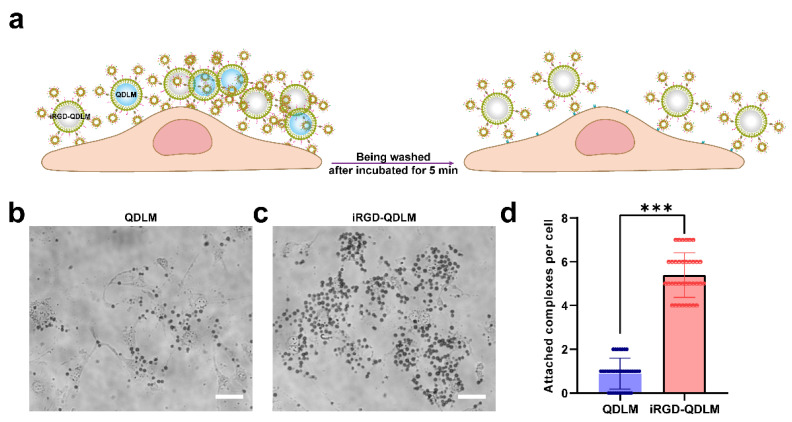
Binding specificity of iRGD-QDLM. (**a**) Schematic diagram of iRGD-QDLM attached to the surface of 4T1 breast cancer cells. (**b**) Representative bright field imaging of QDLM and iRGD-QDLM (**c**) attached to 4T1 breast cancer cells. Scale bar: 50 μm. (**d**) Quantitative analysis of the number of attached complexes per cell. *** *p* < 0.001.

**Figure 4 pharmaceutics-14-02510-f004:**
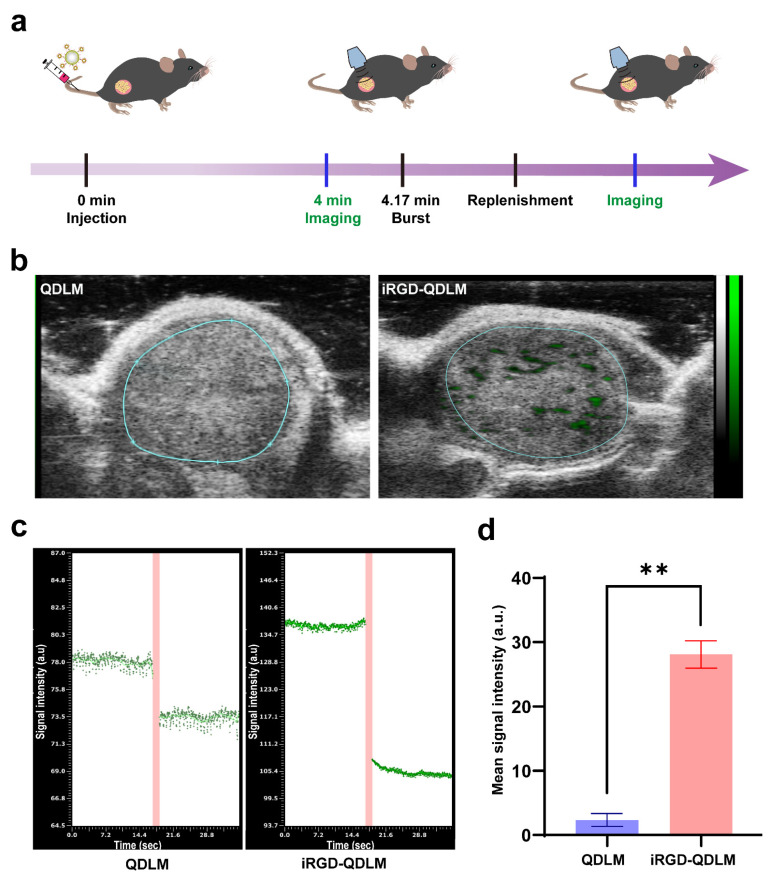
In vivo ultrasound molecular imaging of iRGD-QDLM in the tumor blood vessels. (**a**) Schematic diagram of experimental procedure in a tumor-bearing mouse. (**b**) B-mode image with overlaid non-targeted (QDLM) and αvβ3-integrin-targeted (iRGD-QDLM) ultrasound molecular imaging signals. Green signals represent iRGD-QDLM attached to the tumor vessel. *n* = 5 mice in both iRGD-QDLM and QDLM groups. (**c**) Representative time–signal intensity curves for the region of interest after injection of QDLM and iRGD-QDLM. (**d**) Quantitative analysis of the difference signal intensities of the QDLM and iRGD-QDLM by a destruction/replenishment approach. ** *p* < 0.01.

**Figure 5 pharmaceutics-14-02510-f005:**
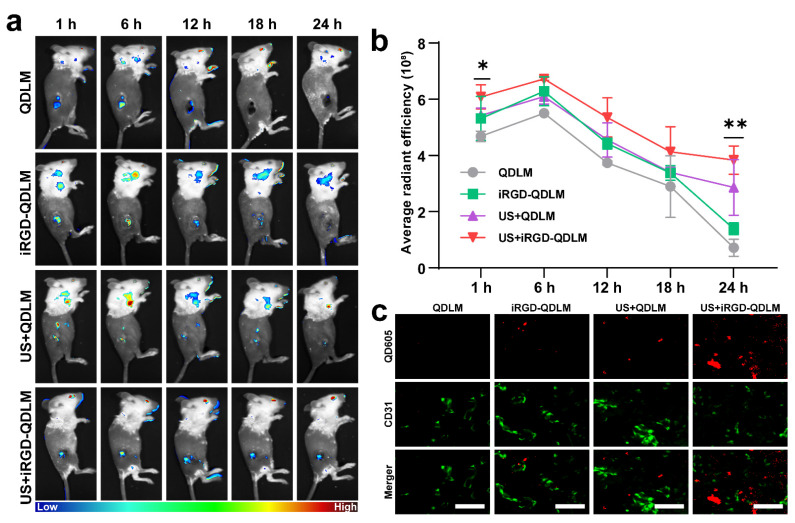
In vivo transvascular fluorescent imaging of iRGD-QDLM in the tumors. (**a**) In vivo fluorescent imaging of the tumor-bearing mice treated with QDLM, iRGD-QDLM, US + QDLM, and US + iRGD-QDLM. *n* = 5 mice in all groups. (**b**) Quantitative analysis of the average signal intensities in the brain after treatment with QDLM, iRGD-QDLM, US + QDLM, and US + iRGD-QDLM at different times. * *p* < 0.05 vs. control, ** *p* < 0.01 vs. control. (**c**) Confocal images of the tumor slices stained by anti-CD31 (green) after 24 h post treatment of QDLM, iRGD-QDLM, US + QDLM, and US + iRGD-QDLM. Red color represents QD605. Scale bar: 20 µm.

## Data Availability

Not applicable.

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
