# Peer review of "Dual-Modality Molecular Imaging of Tumor via Quantum Dots-Liposome–Microbubble Complexes"

_pharmaceutics, 2022, doi:10.3390/pharmaceutics14112510_

Round 1

Reviewer 1 Report

Wang et al. presented dual-modality imaging of breast cancer in cell culture, and mouse models using Qunatumdot-liposomes fused microbubbles. This reviewer encourages the study and recommends publication after a few minor corrections. 

Minor points:

1. Methodology or ratios of materials used in the preparation of microbubbles is not clear in the methods section.  

2. This current study is interesting and can be impactful if authors use any current state-of-art method to compare and benchmark the performance of their method. As this is beyond the scope of this article, the authors should explain the possible advantages of their method by comparing current clinical practices.  

Author Response

Reviewer #1:

Wang et al. presented dual-modality imaging of breast cancer in cell culture, and mouse models using Qunatumdot-liposomes fused microbubbles. This reviewer encourages the study and recommends publication after a few minor corrections. 

Response: Thank you very much for your constructive suggestions. We have made some revision to make it more reasonable. The detailed changes can be found in the revised manuscript.

Comment 1: Methodology or ratios of materials used in the preparation of microbubbles is not clear in the methods section.  

Response: We have added ratios of materials used in the preparation of microbubbles in the methodology.

“Briefly, appropriate proportion of DSPC, DSPE-PEG2000 and DSPE-PEG2000-Biotins (molar ratio, 90:5:5) were dissolved in 1mL chloroform.”

Comment 2: This current study is interesting and can be impactful if authors use any current state-of-art method to compare and benchmark the performance of their method. As this is beyond the scope of this article, the authors should explain the possible advantages of their method by comparing current clinical practices.  

Response: We have demonstrated the advantages of our dual-modality imaging in the discussion by comparing current clinical practices.

“Compared with molecular-imaging modalities used in clinical practice (MRI, CT, PET and SPECT), our dual-modality imaging integrates the advantages of non-invasiveness and relative tolerability of the ultrasound imaging and the properties of high sensibility of optical imaging. Actually, the clinical imaging modalities suffer from major limitations including adverse effects to hazardous ionizing radiation (CT, PET and SPECT), intrinsi-cally limited spatial and poor temporal resolutions (CT, MRI, PET and SPECT). More im-portantly, novel dual-modality imaging achieved transvascular fluorescent imaging via ultrasound targeted destruction technology.”

Reviewer 2 Report

The paper by Wang, et al. shows the use of quantum dots loaded microbubbles for dual mode imaging of breast cancer tumors. While the results seem promising some more detailed additional experiments and discussion should be added to the manuscript. A main concern is that the paper did not highlight/explain why the specific quantum dot was chosen instead of other more well-known and more biocompatible dyes (e.g. indocyanine green). The authors need to explain to the particular advantage of using this quantum dot compared to what has been reported in literature especially given that the quantum dot was purchased and not synthesized. In addition, the microbubbles were not characterized well with no exact number provided for size and polydispersity and nonlinear contrast imaging was not performed or no images were provided including time intensity curves and stability studies are lacking (this would be possible given that they used a Vevo2100). Kindly see my comments below.

11.       Kindly show in vitro cell toxicity assay of the quantum dots used in this study to prove biocompatibility.

22.       Please explain why other common dyes such as indocyanine green or other quantum dots used in literature was not used instead to provide the optical imaging properties for their agent?

33.       More details about the microbubble properties of the QD-MBs need to be provided:

a.       Provide in vitro and in vivo ultrasound imaging stability studies of the bubbles by providing ultrasound time intensity curves (using nonlinear mode imaging ideally).

b.       Please add exact values for the size and standard deviation of the bubbles. Figure 2b only showed the curve but no exact numbers.

c.       Given that MBs are unstable, please also provided time dependent size and concentration. This way we will know whether the agent is stable and what timeframe it needs to be used after synthesis.

44.       How many cells (n=?) were analyzed for the number of QD-MBs bound to the cells? Imaging only gives a very small area and in Figure 3d it seems only n = 17 cells were analyzed, which is a very small number given the large number of cells used in the binding assay. This should be much higher to be statistically relevant given the initial number of cells.

55.       Do we assume that after washing unbound MBs that all of them adhere to the cells? If yes, please use fluorescence measurement to quantify quantum dots-MBs bound to the cells for more accurate measurements and compare to imageJ analysis as comparison?

66.       How was tumor volume calculated and what were the exact values used in the study. Please provided the number of tumors, the average size, and standard deviation. Always provide exact values that were used in the study.

77.       Was nonlinear contrast mode used for in vivo imaging of the QD-MBs bubbles or only B-mode? Kindly add nonlinear contrast mode, which will further highlight the importance and advantage of using MBs.

Author Response

Reviewer #2:

The paper by Wang, et al. shows the use of quantum dots loaded microbubbles for dual mode imaging of breast cancer tumors. While the results seem promising some more detailed additional experiments and discussion should be added to the manuscript. A main concern is that the paper did not highlight/explain why the specific quantum dot was chosen instead of other more well-known and more biocompatible dyes (e.g. indocyanine green). The authors need to explain to the particular advantage of using this quantum dot compared to what has been reported in literature especially given that the quantum dot was purchased and not synthesized. In addition, the microbubbles were not characterized well with no exact number provided for size and polydispersity and nonlinear contrast imaging was not performed or no images were provided including time intensity curves and stability studies are lacking (this would be possible given that they used a Vevo2100). Kindly see my comments below.

Response: According to reviewer's kindly suggestions, we have demonstrated particular advantage of this quantum dot in the revised manuscript. Traditionally, for various bioimaging applications, fluorophores such as indocyanine green are widely accepted. Although the fluorophores are preferred, but they are susceptible to photobleaching, low signal intensity, and spectral overlapping. In case of organic dyes, the microenvironment (including matrix polarity, viscosity, pH and ionic strength and presence of surfactants) affects the spectral position, fluorescence lifetime and the intensity of the absorption and emission bands. These limitations hinder their use especially in multiplexed bioimaging applications. QDs are better in the case of multispectral imaging as they exhibit size tunability with narrow emission spectrum. Moreover, they are thermally and photo-chemically stable as compared to their counterparts. Furthermore, they exhibit a broad range of emission wavelengths which extend deep into the NIR region with longer fluorescence lifetime. However, QDs need to be biocompatible and reproducible to be suited as an ideal bioimaging agent. Therefore, QDs were encapsulated into liposome for good biocompatibility in our study.

Comment 1: Kindly show in vitro cell toxicity assay of the quantum dots used in this study to prove biocompatibility.

Response: Thank you for you constructive suggestions, we have supplemented in vitro cell toxicity assay of the quantum dots. As shown in Figure I, different concentrations of iRGD-QDLM and QDLM all had no effect on 4T1 cell viability. In addition, high concentrations of quantum dots influenced the cell viability to some extent. However, quantum dots encapsulated into liposome would reduce its cytotoxicity and improve its biocompatibility. In addition, concentrations of quantum dots were usually lower than 1 mg in the experiment.

Figure I. Cell viability assay of of 4T1 cells determined by the CCK-8 assay after incubation with various concentrations of quantum dots (QD), αvβ3 integrin-targeted quantum dots loaded liposome-microbubbles complexes (iRGD-QDLM), and quantum dots loaded liposome-microbubbles complexes (QDLM) for 24 h.

Comment 2: Please explain why other common dyes such as indocyanine green or other quantum dots used in literature was not used instead to provide the optical imaging properties for their agent?

Response: QD605 provides the following advantages: (I) It was better in the case of multispectral imaging as they exhibit size tunability with narrow emission spectrum; (II) It was thermally and photo-chemically stable as compared to their counterparts; (III) It exhibited a broad range of emission wavelengths which extend deep into the NIR region with longer fluorescence lifetime; (IV) It had good optical imaging properties both in vivo and in vitro.

Comment 3: More details about the microbubble properties of the QD-MBs need to be provided:

Response:  We have supplemented more details about the microbubble properties of the QD-MBs (Table 1). In addition, we supplemented in vitro and in vivo ultrasound imaging stability and time dependent size and concentration of the QD-MBs.

Table 1 The properties of the QD-MBs.

Samples

QDLM

iRGD-QDLM

Concentration/107#/mL

1.31 ± 0.17

1.59 ± 0.13

Particle size/μm

1.56 ± 0.92

1.67 ± 1.11

Ultrasound Images

Echo-Power/A.U.

116.63 ± 23.5

114.46 ± 13.5

Comment 4: Provide in vitro and in vivo ultrasound imaging stability studies of the bubbles by providing ultrasound time intensity curves (using nonlinear mode imaging ideally).

Response: We have supplemented in vitro and in vivo ultrasound imaging stability studies of iRGD-QDLM.  in vitro and in vivo ultrasound images of iRGD-QDLM were taken at different time points. Figure II demonstrated  that  ultrasound imaging signals of iRGD-QDLM decreased over time. However, ultrasound imaging signals of iRGD-QDLM remained unchanged during the first 5 minutes (Figure III), which paved the way for a destruction/replenishment experiment.

Figure II. In vitro and in vivo ultrasound imaging of iRGD-QDLM at different time.

Figure III.Time intensity curves of the acoustic imaging signal in vitro and in vivo ultrasound imaging of iRGD-QDLM at different time.

Comment 5: Please add exact values for and standard deviation of the bubbles. Figure 2b only showed the curve but no exact numbers.

Response: We added exact values for the size and standard deviation of the bubbles in the revised manuscript. The size of iRGD-QDLM was1.56 ± 0.92μm and The size of QDLM was 1.67 ± 1.11μm.

Comment 6: Given that MBs are unstable, please also provided time dependent size and concentration. This way we will know whether the agent is stable and what timeframe it needs to be used after synthesis.

Response: To further demonstrate the stability MBs, time dependent size and concentration of MBs were analyzed. Time-concentration and -size curve showed that both solutions remained a high concentration and stable size of at least 15 min (Figure IV, V). Therefore, prepared iRGD-QDLM should be used within 15 minutes.

Figure IV. Size-time curve.

Figure V. Concentration-time curve.

Comment 7: How many cells (n=?) were analyzed for the number of QD-MBs bound to the cells? Imaging only gives a very small area and in Figure 3d it seems only n = 17 cells were analyzed, which is a very small number given the large number of cells used in the binding assay. This should be much higher to be statistically relevant given the initial number of cells.

Response: Actually, it just was representative bright field imaging of iRGD-QDLM attached to 4T1 breast cancer cells. In addition, more cells were analyzed for the number of QD-MBs bound to the cells in Figure 3d.

Figure VI. Bright field imaging of iRGD-QDLM attached to 4T1 breast cancer cells.

Figure 3 (d) Quantitative analysis of the number of attached complexes per cell. ***P<0.001.

Comment 8: Do we assume that after washing unbound MBs that all of them adhere to the cells? If yes, please use fluorescence measurement to quantify quantum dots-MBs bound to the cells for more accurate measurements and compare to imageJ analysis as comparison?

Response: Actually, residual microbubble shell material and some intact microbubbles could adhere to the cells by electrostatic adsorption, which affected more accurate measurements. Therefore, direct observation of QD-MBs bound to the cells proved valuable for analysis of targeted QD-MBs. Figure 3b demonstrated that large amounts of iRGD-QDLM were attached to the surface of 4T1 breast cancer cells, proving the specific attachment between iRGD- iRGD-QDLM and breast cancer cells.

Comment 9: How was tumor volume calculated and what were the exact values used in the study. Please provided the number of tumors, the average size, and standard deviation. Always provide exact values that were used in the study.

Response: The tumor volume (mm3) was calculated using the formula (length × width × height) × 0.5 in millimeters. The size of the tumors was monitored and permitted to grow to a size of 6–10 mm in diameter over two weeks. After the tumor volume reached an average size of 150.3±33.1 mm3, all of the experimental animals were randomly assigned to different groups in the following experiments.

“Tumor volume (V) was calculated according to the formula: V = ((length (mm) × width 2 (mm)) × 0.5). When tumour volume was 150.3 ± 33.1 mm3, tumor-bearing mice on a heat-ing pad were anesthetized with 2% isoflurane.”

“n = 5 mice in both iRGD-QDLM and QDLM groups.”

“n = 5 mice in all groups.”

Comment 10: Was nonlinear contrast mode used for in vivo imaging of the QD-MBs bubbles or only B-mode? Kindly add nonlinear contrast mode, which will further highlight the importance and advantage of using MBs.

 Response: We have supplemented nonlinear contrast mode used for in vivo imaging of the iRGD-QDLM and QDLM. As shown in Figure VII, iRGD-QDLM and QDLM produce intense nonlinear ultrasound signals inside the tumor. These results further highlighted the importance and advantage of using iRGD-QDLM.

Figure VII. In vivo acoustic imaging of the tumor was taken after tail vein injection of iRGD-QDLM or QDLM.
